# Targeted Analysis of the Gut Microbiome for Diagnosis, Prognosis and Treatment Individualization in Pediatric Inflammatory Bowel Disease

**DOI:** 10.3390/microorganisms10071273

**Published:** 2022-06-22

**Authors:** Christine Olbjørn, Milada Cvancarova Småstuen, Aina Elisabeth Fossum Moen

**Affiliations:** 1Department for Paediatric and Adolescent Medicine, Akershus University Hospital, NO-1470 Lørenskog, Norway; 2Department of Nursing and Health Promotion, Oslo Metropolitan University, NO-0176 Oslo, Norway; miladacv@medisin.uio.no; 3Division of Infection Control and Environmental Health, Norwegian Institute of Public Health, NO-0213 Oslo, Norway; aefm@fhi.no

**Keywords:** IBD, microbiota, pediatric, prognosis, biologic therapy, biomarker

## Abstract

We explored the fecal microbiota in pediatric patients <18 years of age with treatment-naïve IBD (80 Crohn’s disease (CD), 27 ulcerative colitis (UC)), in 50 non-IBD patients with gastrointestinal symptoms without inflammation and in 75 healthy children. Using a targeted qPCR approach, the quantities of more than 100 different bacterial species were measured. **Results:** The bacterial abundance was statistically significantly reduced in the IBD and non-IBD patients compared to the healthy children for several beneficial species. The CD patients had a lower abundance of *Bifidobacterium* species compared to the UC patients, and the IBD patients in need of biologic therapy had a lower abundance of butyrate producing bacteria. Based on the abundance of bacterial species at diagnosis, we constructed Diagnostic, Phenotype and Prognostic Indexes. Patients with a high Diagnostic Index had 2.5 times higher odds for having IBD than those with a lower index. The CD patients had a higher Phenotype Index than the UC patients. Patients with a high Prognostic Index had 2.1 higher odds for needing biologic therapy compared to those with a lower index. **Conclusions:** The fecal abundance of bacterial species can aid in diagnosing IBD, in distinguishing CD from UC and in identifying children with IBD in need of biologic therapy.

## 1. Introduction

Inflammatory bowel diseases (IBD)—Crohn’s disease (CD) and ulcerative colitis (UC)— are chronic, lifelong inflammatory diseases of the gastrointestinal (GI) tract. The pathogenesis is not fully understood, and there are no reliable biomarkers to diagnose and predict the clinical course. Disturbances in the gut microbiota are thought to be important in the development of IBD [1], and a leading hypothesis is that the intestinal inflammation is caused by an inappropriate immune response to commensal bacteria in genetically susceptible individuals [2]. Studies of the gut microbiota in IBD patients have shown an imbalance, dysbiosis, with compositional changes, including decreased bacterial diversity and abundance [3,4]. Dysbiosis is also linked to a broad spectrum of gastrointestinal conditions besides IBD [5] such as irritable bowel syndrome (IBS) [6]. Symptoms in IBD patients are often non-specific, with many of the symptoms of IBD being present in patients who do not have gut inflammation and, hence, need other types of treatment. Differentiating IBD patients from this group can be challenging [7]. Diagnosing IBD in children warrants upper and lower endoscopies in general anesthesia, making the diagnostic work-up invasive and resource intensive. Diagnostic delays are common [8]. In pediatric IBD, it may be difficult to categorize the patient’s phenotype correctly. In younger children, CD often has a colonic distribution, making the differentiation to UC challenging, as the ileal involvement in CD, often present in adults, occurs at a later age [9,10]. It is important to distinguish whether the patient has CD or UC, as the treatment is different, with the use of nutritional therapy in CD, such as exclusive enteral nutrition [11]. The individual disease course of IBD is unpredictable, but children with IBD are more frequently in need of immunosuppressive and biologic therapy, as pediatric IBD is characterized by an extensive disease distribution and an aggressive disease course coinciding with the onset of puberty [12,13,14]. Early diagnosis and effective therapy to avoid irreversible complications and delayed growth are vital. Despite numerous studies during the last decade using the gut microbiome and 16S rRNA sequencing to diagnose IBD and predict treatment escalation, predictive values have been below the levels required [15]. More knowledge on etiological and prognostic factors is needed to increase the diagnostic precision and tailoring of treatments.

Our aim was to explore whether a novel qPCR-based targeted approach, providing absolute bacterial quantification down to a species and sub-species level, could discriminate patients with IBD from patients with symptoms similar to IBD but without inflammation.

We also wanted to explore whether bacterial species were associated with the phenotype and treatment needed in the first years after the diagnosis of IBD. Further, we propose indexes based on the detected differences in fecal microbiota species abundance, which might aid in the identification of patient subgroups.

## 2. Materials and Methods

Our patients, all under 18 years, were recruited from the catchment areas of two university hospitals in three population-based prospective epidemiological studies of treatment-naïve pediatric IBD in South-Eastern Norway (IBSEN II, Early IBD and EU IBD Character) [13,16,17,18]. The inclusion periods for these three multicenter trials were from 2005 to 2015, all with identical protocols and inclusion criteria. Pediatric patients who were referred during the inclusion periods and were believed to have IBD based on symptoms were included. IBD was diagnosed in accordance with the Porto criteria [14]. Patients who did not meet the diagnostic criteria for IBD and who had a macroscopically and histologically normal mucosa as well as a normal MRI examination were included as non-IBD symptomatic controls. Healthy children and adolescents between the age of 2 and 18 years, recruited during the period of 2013–14 from the same catchment areas as the patients, delivered fecal samples and were included as healthy controls. They had no chronic diseases, no IBD in the family, followed a normal diet (children on exclusion diets—gluten-free, cow’s milk protein-free, vegetarian/vegan—were excluded), had not travelled outside Europe or used antibiotics within the last six months, had no recorded gastrointestinal complaints, did not use proton pump inhibitors, did not smoke and had normal fecal calprotectin levels (<50 mg/kg).

### 2.1. Clinical, Endoscopic, Radiological and Laboratory Data

The age, gender, symptoms, disease activity index scores, disease and family histories of the IBD and non-IBD symptomatic patients were registered as previously described [13,17,18]. The Paris classification was used to characterize disease distribution and behavior [19]. All patients were examined with upper and lower endoscopies with biopsies from all parts of the GI tract. Magnetic resonance imaging (MRI) studies were performed to examine the small bowel. In the patients, the feces were sampled at home in three designated containers without additives on the day before endoscopy, kept refrigerated or frozen and brought to the hospital the next day. The feces from one container was analyzed for calprotectin (FeCal-test, Bühlmann, Basel, Switzerland), the second was analyzed for pathogenic bacteria and the third container with feces was frozen at −80 degrees Celsius for later microbiota analysis. The healthy controls received two designated fecal sampling kits at home for the handling of samples. One sample was analyzed for fecal calprotectin (FeCal-test, Bühlmann, Basel, Switzerland), the other was frozen at −80 degrees Celsius and stored for later microbiota analysis. For all samples, the maximum time interval until freezing at −80 degrees Celsius was three days; thereafter, the samples were kept frozen and not thawed until analysis. DNA purification from the fecal samples was performed as described by Casén et al. [20].

### 2.2. IBD Treatment

The treatment was decided individually at the discretion of the treating pediatrician. The initial treatment options to induce remission were: exclusive enteral nutrition (EEN) in CD and corticosteroids and/or 5-aminosalicylic acids in CD and UC patients. Maintenance therapy with azathioprine or methotrexate was generally started simultaneously. The indication for surgery or treatment with biologic therapy (tumor necrosis factor (TNF) blockers) was the failure to induce remission with conventional treatments or relapse after primary induction.

### 2.3. Microbiota Analysis

We determined the bacterial taxonomic composition of the fecal microbiota using a targeted qPCR approach to microbiome profiling (the PMP™ technology platform from Bio-Me, Oslo, Norway). PMP™ gives an absolute quantification down to the species and sub-species level of the most dominant, frequent and relevant bacteria in the sample.

The PMP™ panel utilizes the OpenArray^®^ technology from ThermoFisher Scientific (ThermoFisher Scientific, Waltham, MA, USA) and is able to quantify the presence of more than 100 different bacterial species and sub-species in a sample [21]. We used 107 bacterial targets for the first panel, adding additional 47 targets to the extended panels for a subset of analyses (Appendix A). The OpenArray^®^ panels were run on the QuantStudio™ 12 K qPCR platform (ThermoFisher Scientific). The liquid handling steps were automated and performed using the epMotion™ 5700 (Eppendorf, Hamburg, Germany) and Accufill™ systems (ThermoFisher Scientific). The absolute quantification of the number of genomic copies per μL for each bacterial taxon was interpolated from standard curves derived from quantified reference isolates (Appendix A). In our analyses, we used the relative abundance (%), which is the total number of copies for a given target divided by the sum of copies for all target bacteria included in the PMP™ report.

### 2.4. Statistical Analyses

The resulting data were described using counts and percentages for categorical data and medians and ranges for continuous data. Crude comparisons between the groups were performed using the Mann–Whitney and Wilcoxon signed ranks tests (before and after treatment) for continuous variables and Chi-square tests for categorical data. The Areas Under the Curves (AUCs) and Area Under Precision-Recall Curves (AUPRCs) were calculated, and receiver operating characteristic (ROC) analysis was conducted to evaluate the performance of selected bacterial abundances in distinguishing the different diagnoses, subgroups based on phenotypes and treatments. Based on the detected differences in fecal microbiota between the groups, we constructed five indexes. Diagnostic Indexes, quantifying the likelihood of having IBD versus healthy (Diagnostic Index 1), non-IBD versus healthy (Diagnostic Index 2) and IBD versus non-IBD (Diagnostic Index 3); the Phenotype Index for CD versus UC; and the Prognostic Index, quantifying the likelihood of receiving biologic therapy versus conventional therapy based on the microbiota abundance at diagnosis. First, we computed the median values for the bacteria, which were significantly different between the groups selected for each index. An individual scored 1 if his/her values were higher than the median for the comparable group for each of the bacteria. The index was made by summing up the values for each categorized bacterium. Further, we added weights to those bacteria with the most diverting abundances, thus giving more “power” to the bacteria for which inter-group differences were the most pronounced. Lastly, we fitted logistic regression models with the indexes as the dependent variables. The results are expressed as odds ratios (OR) with 95% confidence intervals (CI). Given the limited sample size, the CIs were constructed using bootstrapping with 10,000 repetitions. The performance of each model was evaluated using leave-one-out cross-validation (LOOCV), and the results are reported as the mean squared error (MSE), calculated as the average of all MSEs for each model with one observation left out. All tests were two-sided. *p*-values < 0.05 were considered statistically significant. We regarded our study exploratory; therefore, we did not correct for multiple testing. All analyses were performed using SPSS, statistical software version 24 (SPSS Inc., Chicago, IL, USA) and Stata version 17.

## 3. Results

Of the 235 included children and adolescents, IBD was diagnosed in 110 patients, (80 CD, 27 UC and 3 IBD unclassified), 50 patients were included as non-IBD symptomatic patients and 75 healthy children served as controls (Table 1). None of the non-IBD symptomatic patients have developed IBD as of 1 December 2021. The IBD, non-IBD and healthy controls were comparable concerning all the presented demographic variables except for more females among the non-IBD patients and a slightly lower median age in the healthy controls (Table 1). There were no statistically significant differences in the microbiota abundances regarding age or sex, and all of the patients were treatment-naïve at the time of microbiota sampling.

### 3.1. Phenotypes

Most of the CD patients had an inflammatory phenotype—53 (66%)—while 13 (16%) had stricturing and 14 (18%) had penetrating phenotypes. Ileocolonic involvement was the most common, with 47 (59%), followed by colonic involvement in 24 (30%) and isolated ileal disease in 5 (6%) patients. Seventeen (21%) had perianal involvement and fifty-four (68%) had upper gastrointestinal involvement. Of the UC patients, 17 (63%) had total colitis.

### 3.2. Treatment of IBD Patients

Of the IBD patients, 64 (58%) were later treated with biologic therapy, while the rest were treated with conventional mediation—5-amino salicylic acid (5-ASAs), corticosteroids and azathioprine in UC and exclusive enteral nutrition and immunomodulators (azathioprine and methotrexate) in CD. During follow-up, (range 5–18 years), 22 (34%) of the patients receiving biologic therapy needed treatment escalation to a second biologic drug due to non-response, antibody formation to infliximab or a loss of response. A total of 17 (15%) of the IBD patients underwent surgery.

### 3.3. Microbiota in IBD, Non-IBD and Healthy

The bacterial abundance was significantly reduced in the patients (IBD and non-IBD grouped) compared to the healthy controls for several species. Comparing the non-IBD patients to the healthy controls revealed 14 species with reduced abundances in the non-IBD patients (Table 2). The IBD patients had significantly reduced abundance for 30 bacterial species compared to the healthy controls and reduced abundance of 21 bacterial species compared with the non-IBD patients (Table 2).

### 3.4. Microbiota in IBD Patients and Association with Phenotypes and Treatment

The CD patients had a significantly lower abundance for five bacterial species and a higher abundance for five bacterial species. The CD patients with stricturing and/or penetrating phenotypes had a lower abundance of four species—namely, *Christensinella minuta* (*p* = 0.046), *Clostridium scindens* (*p* = 0.027), *Eubacterium eligens* (*p* = 0.047) and *Roseburia hominis* (*p* = 0.05)—and a higher abundance of *Escherichia coli* (*p* = 0.010) compared to the CD patients with an inflammatory phenotype.

The IBD patients receiving biologic therapy had a significantly lower abundance of ten bacterial species and a higher abundance of two bacterial species compared to the conventionally treated patients (Table 3). A subset of the IBD patients receiving biologic therapy needed further treatment escalation to a second class of biologics. These patients had a significantly lower abundance of *Bifidobacterium bifidum*, *Roseburia hominis* and *Bacteroides xylanisolvens* compared to the IBD patients who received one biologic or conventional therapy.

### 3.5. Microbiota and Association with Fecal Calprotectin

The IBD patients with higher fecal calprotectin levels (31 with CD and 12 with UC had above 1000 mg/kg) had a significantly lower abundance of five bacterial species compared to the patients with lower levels of fecal calprotectin (Table 4). Fecal calprotectin over 1000 mg/kg was associated with subsequent biologic therapy, *p* = 0.001, but not with later surgery.

### 3.6. Indexes: Diagnostic, Phenotype and Prognostic Index

Based on the detected differences in microbiota abundance between the pairs of selected groups (IBD patients and non-IBD patients versus healthy individuals, IBD versus non-IBD, CD vs. UC patients and IBD patients treated with biologic therapy versus conventional medication), we constructed indexes where higher scores indicated a higher likelihood of belonging to one of the groups. Some bacterial species were included in several of the constructed indexes (see Table 4). To quantify this likelihood, we fitted separate logistic regression models and computed the odds for having a given outcome for each unit increase of a given index.

### 3.7. The Diagnostic Index 1 (IBD Patients vs. Healthy Individuals)

For each unit increase of the Diagnostic Index (DI) 1, the odds for having IBD vs. being healthy, there was more than a twofold increase in the odds for having IBD (OR = 1.33 95%CI [1.22 to 1.46], *p* < 0.001. The DI1 had good discrimination properties, with an AUC of 0.78, 95%CI [0.72 to 0.85] (Figure 1a,b). The Area Under Precision-Recall Curves (AUPRCs) for the DI1 are available as Appendix A.

### 3.8. The Diagnostic Index 2 (Non-IBD Patients vs. Healthy Individuals)

For each unit increase of the DI2, the odds for having non-IBD vs. being healthy, there was more than a twofold increase in the odds for having non-IBD (OR = 2.23; 95%CI [1.57 to 3.88], *p* = 0.001). The DI2 had good discrimination properties, with an AUC of 0.77, 95%CI [0.68 to 0.86] (Figure 2a,b). The AUPRC for the DI2 is available as Appendix A.

### 3.9. The Diagnostic Index 3 (IBD Patients vs. Non-IBD Patients)

By testing all of the patients (110 IBD and 50 non-IBD) with 106 probes, we have constructed the DI3, aiming to distinguish between IBD versus non-IBD patients. The DI3 had good discrimination properties, with an AUC of 0.69; 95%CI [0.60 to 0.78]. When testing a subset of patients with spare DNA for 22 additional targets containing bacterial species thought to be important for IBD and gut health, the DI 3 improved, with an AUC of 0.83, 95%CI [0.74 to 0.93]. Fitting a logistic regression model, the patients with a higher DI 3 had 2.5 times higher odds for having IBD compared to the patients with lower scores (OR = 2.55, 95%CI [1.71 to 5.89], *p* < 0.001 (Figure 3a,b). The AUPRC for the DI3 is available as Appendix A.

### 3.10. Phenotype Index (CD Patients vs. UC Patients)

A higher Phenotype Index (PhI) score indicated a higher likelihood that a patient had CD and not UC. For each unit increase of the PhI, the odds for having CD increased by about 46% (OR = 1.46; 95%CI [1.20 to 2.02], *p* = 0.001. The PhI had good discrimination properties, with an AUC of 0.74, 95%CI [0.64 to 0.85] (Figure 4a,b). The AUPRC for the PhI is available as Appendix A.

#### Prognostic Index (Need for Biologic Therapy vs. No Need)

A higher Prognostic Index (PI) score indicated a need for biologic therapy. For each unit increase of this index, there was a twofold increase in the odds for biologic treatment (OR = 2.1, 95%CI [1.40 to 3.46], *p* = 0.001). The PI had good discrimination properties, with an AUC of 0.72, 95%CI [0.63 to 0.82] (Figure 5a,b). The AUPRC for the PI is available as Appendix A.

## 4. Discussion

The present study adds new knowledge of the gut microbiota as a biomarker to aid in diagnosing children with IBD, distinguishing CD from UC and predicting treatment with biologic therapy and treatment escalation. We show reduced bacterial abundances in IBD and non-IBD patients compared to healthy children for several beneficial microbial species measured with the PMP™ precision platform. The IBD patients had a lower bacterial abundance than the non-IBD symptomatic patients, and bacterial abundances at baseline were associated with disease phenotype (CD versus UC), inflammatory versus stricturing/penetrating phenotypes in CD and a later need for biologic therapy in the IBD patients. The positive relationship between inflammation, an increased abundance of pathobionts and a loss of beneficial bacteria is in line with previous research reports [4,22].

### 4.1. Diagnostic Potential of the Gut Microbiome

The median age for the debut of IBD in children is 12 years, which coincides with the onset of puberty and its associated growth spurt [23]. To avoid irreversible complications and reduced growth, it is important to diagnose IBD and its correct phenotype without diagnostic delay and to start the necessary treatment that will induce remission. We found that several bacterial species at the disease onset and before the initiation of treatment were associated with different phenotypes and later disease severity. The diagnostic and prognostic indexes we constructed could be of aid in diagnosing IBD and in the initial decision making regarding medical therapy.

Previously, we reported the relative bacterial abundance of bacterial DNA markers with the GA map test^®^ in the same patients as those included in the present study [18]. The GAmap^®^ method measures 300 bacterial probes at different taxonomic levels, 19 of them species-specific, compared to up to 150 bacterial species with the PMP™ platform. With the GAmap^®^ approach, we were not able to distinguish CD from UC, and the non-IBD patients had a comparable dysbiosis to that of the IBD patients. With the targeted species approach in the present study, we were able to make five indexes. Three Diagnostic Indexes, quantifying the likelihood of having IBD, non-IBD or being healthy; a Phenotype Index, quantifying the likelihood of having CD versus UC; and the Prognostic Index, quantifying the likelihood of needing biologic therapy based on the bacterial abundance at diagnosis. It seems that including more species in the analyses may provide better accuracy and discriminating abilities.

There have been conflicting reports as to how stool samples perform in classifying bacterial dysbiosis compared to mucosal samples. In contrast to the Gevers study, which reported that fecal microbiota samples performed less well than mucosal-associated microbiota samples in pediatric CD [24], we found that fecal microbiota species can discriminate CD from UC as well as IBD from non-IBD symptomatic children. Several other studies have shown altered fecal and mucosal microbiota in both pediatric IBD [25] and IBS patients [26], with promising results with regard to the fecal microbiota adding value in differentiating pediatric IBD patients from controls. In addition to the importance of establishing whether fecal microbiota can contribute to early diagnosis and information on the treatment response, it is important to establish whether fecal microbiota testing is comparable to tests on mucosal-associated microbiota regarding diagnostic accuracy. Sampling fecal microbiota is non-invasive and is easier to assess than mucosa-associated microbiota, which must be obtained during colonoscopy in general anesthesia.

With the PMP™ panel, we found that the non-IBD patients had different bacterial abundances for several species compared to both the healthy children and IBD patients. This finding could serve as a biomarker for an unhealthy gut, as the non-IBD patients otherwise do not have objective findings that correlate with their clinical symptoms. We calculated Diagnostic Indexes, giving the likelihood of having non-IBD versus being healthy and having non IBD versus IBD. A negative fecal calprotectin used together with a Diagnostic Index for the non-IBD symptomatic patients might reduce the need for further invasive investigations for this big group of patients.

### 4.2. Loss of Beneficial Microbes

In line with previous reports, the present study reveals that IBD is predominantly associated with a loss of presumably “beneficial” microbes rather than the introduction of specific pathogens [27]. Mostly, we found reduced abundances of beneficial butyrate-producing bacterial species, which are of importance for gut health. Of special interest was the reduced abundance of the *Bifidobacterium* species *B. bifidum* and *B. adolescentis* in the IBD patients, most pronounced in the CD patients and in the IBD patients who needed biological therapy. Studies have suggested that *Bifidobacteria* protect the intestinal gut barrier [28], act as anti-inflammatory through the modulation of the host immune response and produce vitamins and short chain fatty acids such as butyrate. Furthermore, the importance of *B. bifidum* as a potential microbial biomarker for IBD has been previously reported, where all studies found the species to have a reduced abundance in the disease [22,29,30,31].

We found a reduced abundance of the species within the gut butyrate-producing *Eubacterium* [22,32]. Several studies have shown that a decreased abundance of the *Eubacterium* species is one of the key hallmarks of gut dysbiosis in IBD [4,33]. A Western diet, with a high intake of animal protein and fat and a low fiber consumption, leads to the depletion of both the *Bifidobacterium* and *Eubacterium* species [34]. The increase of a Western diet coincides with an increasing incidence of pediatric IBD in the Western world and underscores the influence of the environment and diet in the pathogenesis of IBD [35,36].

### 4.3. Prognostic Potential

*Roseburia*, a butyrate producer, was reduced in newly diagnosed children with IBD, and reduced levels of *R. hominis* were associated with stricturing and penetrating CD phenotypes, biological therapy, patients who needed treatment escalation with a second biologic drug and a high fecal calprotectin at diagnosis. Our results indicate that *Roseburia* is important for gut health and hemostasis, as species included in the PMP™ panel belonging to this genus were reduced in abundance in the IBD patients compared to both the healthy group and the non-IBD patient group. Higher abundances of *Roseburia* at baseline have, in several studies, both adult and pediatric, been associated with favorable outcomes of biologic therapies [37,38]. One pediatric study showed that children with a higher baseline abundance of *Roseburia* and *E. rectale* were more responsive to anti-TNF-α treatment [39]. In line with these results, we show that patients with low levels of *Roseburia* and *E. rectale* need biologic therapy and might need further treatment escalation to a second biologic drug, indicating that the bacterial profiles at baseline may be helpful in personalizing treatment by finding the optimal drug for each individual [40].

Looking at the bacteria included in our Prognostic Index, most species were markedly reduced at diagnosis in patients needing biologic therapy: the *Bifidobacteria*, *Eubacteria* and *Roseburia* species in addition to *Bacteria finegoldii* and *B. intestinalis*. The abundance of *B. vulgatus* and *R. gnavus* was higher in the IBD patients needing biologic therapy compared to the ones gaining remission with conventional therapy, both species being putative pathogenic bacteria often found enriched in IBD [41]. *B. vulgatus* has been implicated in gut inflammation, and a study by Schirmer et al. revealed an increased activity of this species in both UC and CD patients, indicating a role in the IBD disease manifestation [42]. *B. vulgatus* has been linked to the pathogenesis of CD and NOD 2 host genetic variants [2,43]. In a study from Hall et al., *R. gnavus* was identified as one of the species dominating the gut microbiome of IBD patients; however, the increased abundance was transient [44]. The reasons for the increase are not known but could be related to the inflammatory status of the colonic tissue. All of our IBD patients were treatment-naïve with ongoing inflammation, with the ones receiving biologic therapy having a high inflammatory burden with high fecal calprotectin values, supporting this theory. *R gnavus* has been reported to be more abundant in pediatric CD patients and was one of the top features in a model using microbial abundances in classifying treatment responses with exclusive enteral nutrition in CD patients [45].

We found that the IBD-associated pathogen *H. parainfluenzae* [46] had a relatively higher abundance in the IBD patients compared to the non-IBD symptomatic patients. Schirmer et al. found in their study of pediatric UC patients that *H. parainfluenzae* constituted one of the most significant changes in abundance linked to disease severity [47]. An increasing abundance over time resulted in a failure to achieve remission, and a decrease was associated with improved disease outcome. In a follow up study, it would be of high interest to follow the change in abundance for this species and of the other pathogens, *R. gnavus* and *B. vulgatus*.

### 4.4. Strengths and Limitations

The strength of the present study is the extensive workup, characterization and classification of our IBD patients. All of the non-IBD symptomatic patients underwent the same procedures as the IBD patients, with upper and lower endoscopies as well as the MRI of the small intestine. Our non-IBD symptomatic patients consisted of pediatric patients admitted to the hospital due to symptoms that led to suspicions of IBD but without evidence of inflammation during the workup. We believe that most of these patients had functional gastrointestinal disorders.

Our study has several limitations. Firstly, the sample size is limited, reducing the statistical power to detect small differences in microbiota composition as statistically significant. The healthy controls were not investigated to the same extent as the patients, as invasive tests in healthy children are considered unethical in Norway. Age, diet and smoking are important confounding factors in regard to microbiota composition. Some of our IBD patients and healthy controls were younger than three years of age, before the time when the microbiota becomes more stable, and this could have influenced the results. Different diets are known to influence the microbiota [48]. Unfortunately, breast-feeding patterns and dietary intake were not mapped during the collection period. We excluded healthy children and patients on exclusion diets. Due to the comparable geographic and cultural backgrounds of the study and control groups, we assume that the two groups were likely to have comparable diet habits and do not believe that the diet of the controls differed from the patient’s diets. None of our adolescents admitted to smoking.

The fecal samples were collected at home and, according to instructions, should be placed cold or frozen until delivered to the hospital. No additives were used; thus, deviation from the instructions, e.g., storage at room temperature over longer periods, could influence the abundance of the bacterial species. At the hospital, all of the samples were kept frozen until DNA extraction, limiting the influence of the repeated thawing and freezing on the bacterial and DNA integrity of the samples.

The mere abundance of bacteria gives no information about the functional importance of the species measured. Bacterial 16 S sequencing and shotgun metagenomic sequencing is the state-of-the-art method to quantify bacterial communities. However, it has practical challenges, such as the level of resolution for 16 S sequencing and the complexity and cost for whole genome shotgun sequencing. In addition, both sequencing methods are only able to provide relative abundance information and not absolute abundance, which can lead to erroneous conclusions [49]. Bio-Me’s PMP™ platform provided us with a commercially available, non-invasive and potentially clinically suitable tool with the capacity to generate results within several hours.

We did not adjust for multiple testing, accepting the risk for accepting false positive associations, as we considered this study to be exploratory. Our results should be confirmed with new analyses in larger datasets. In addition, other modeling techniques such as Random Forest or XG-boost might be utilized in addition to the logistic regression models.

The panel of probes in the PMP™ precision platform was not designed specifically for IBD, and there could be other bacteria not tested in this study that could be more important in pediatric IBD diagnosis and prognosis. Testing a subset of patients with additional probes of bacteria not included in the first PMP™ precision platform improved the diagnostic and prognostic accuracy, indicating improved test results with additional bacteria, indicating the room for further optimization. Our results need to be verified and validated in other patient cohorts.

## 5. Conclusions

The fecal abundance of bacterial species differentiated IBD from non-IBD symptomatic patients and healthy controls and could aid in distinguishing CD from UC as well as in identifying IBD patients at a higher risk for aggressive disease with the need for biologic therapy. The abundance of fecal microbiota species may be of diagnostic value and aid in treatment individualization in pediatric IBD patients.

## Figures and Tables

**Figure 1 microorganisms-10-01273-f001:**
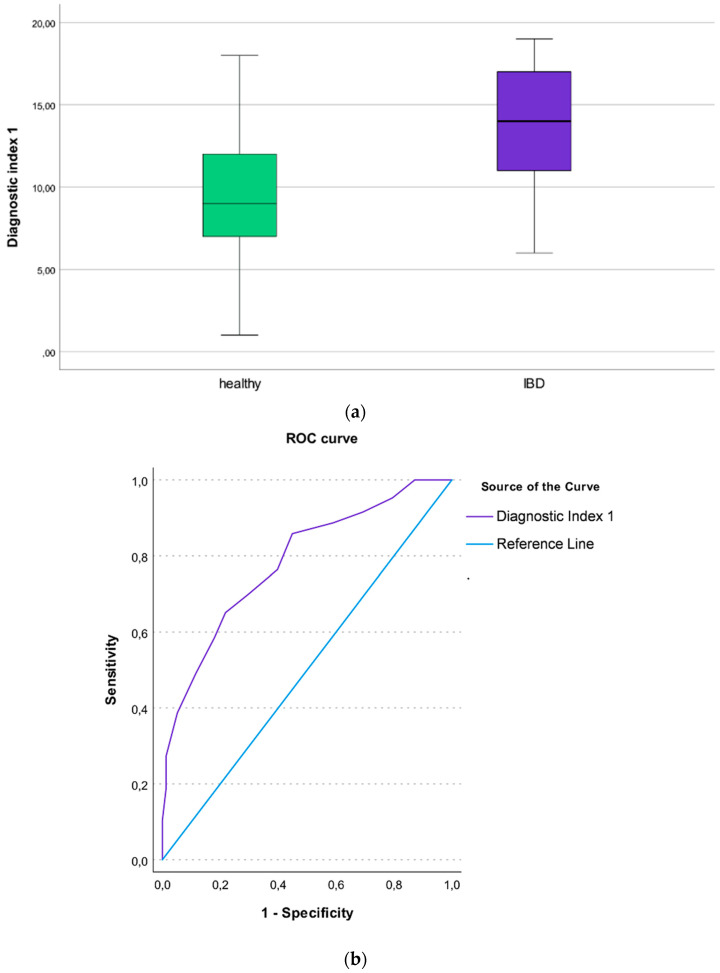
(**a**) Boxplot illustrating the Diagnostic Index 1 differentiating IBD patients from healthy individuals; (**b**) Roc curve showing the AUC of the Diagnostic Index 1 differentiating IBD patients from healthy individuals.

**Figure 2 microorganisms-10-01273-f002:**
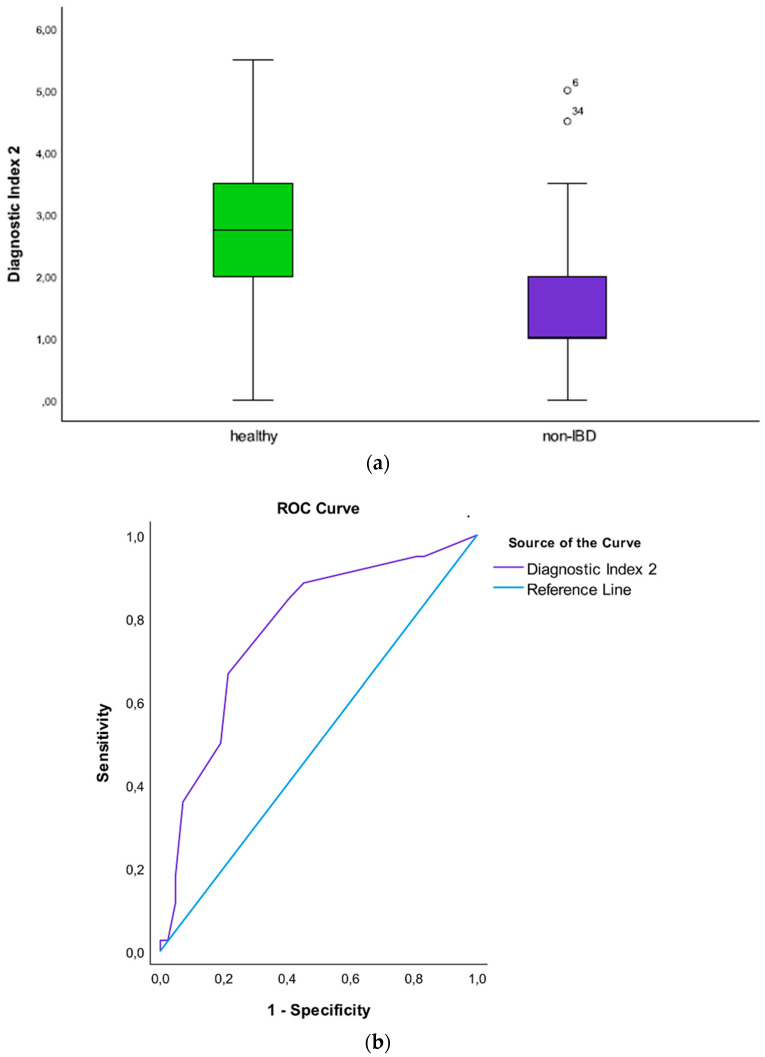
(**a**) Boxplot illustrating the Diagnostic Index 2 differentiating non-IBD patients from healthy individuals; (**b**) Roc curve showing the AUC of the Diagnostic Index 2 differentiating non-IBD patients from healthy individuals.

**Figure 3 microorganisms-10-01273-f003:**
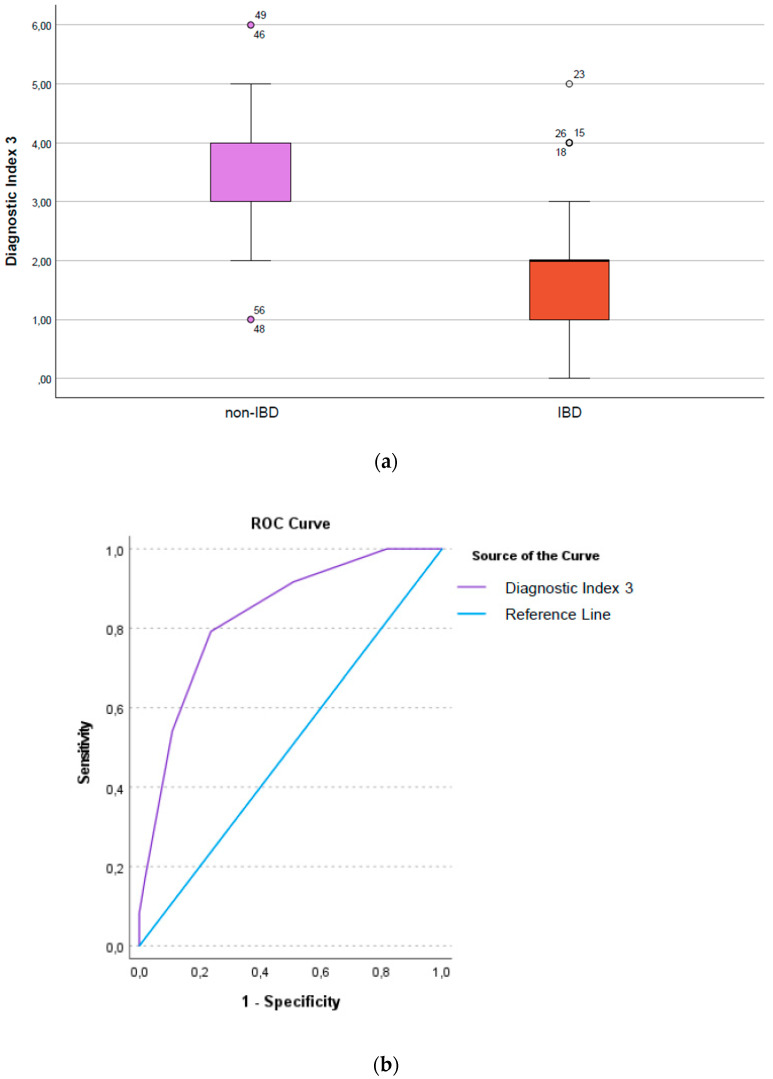
(**a**) Boxplot illustrating the Diagnostic Index3, differentiating non-IBD from IBD patients; (**b**) Roc curve showing the AUC of the Diagnostic Index 3, differentiating non-IBD from IBD patients.

**Figure 4 microorganisms-10-01273-f004:**
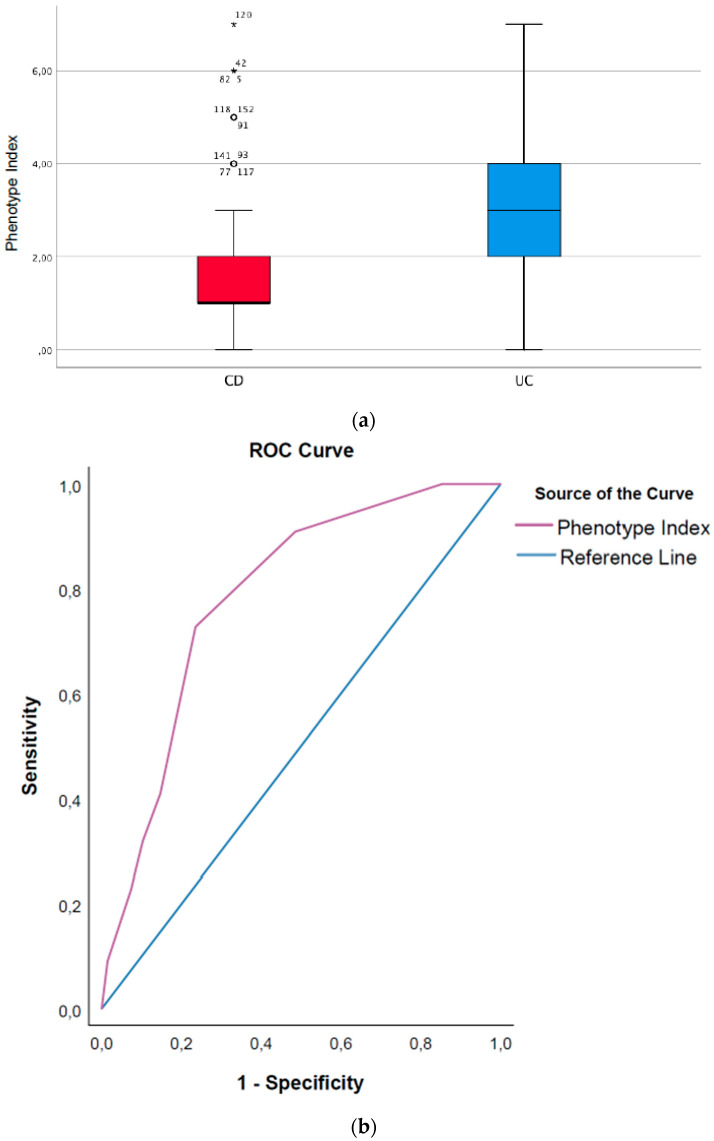
(**a**) Boxplot illustrating the Phenotype Index differentiating between CD and UC; (**b**) Roc curve showing the AUC of the Phenotype Index differentiating between CD and UC.

**Figure 5 microorganisms-10-01273-f005:**
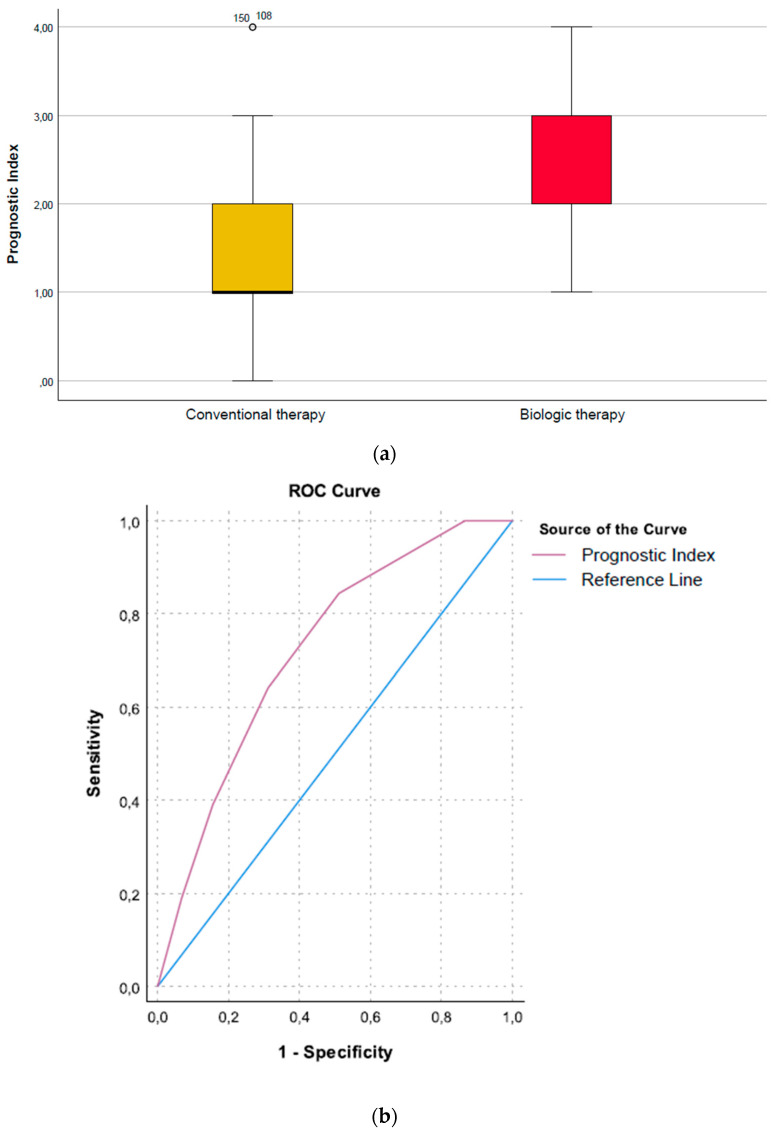
(**a**) Prognostic Index showing the need for biological therapy; (**b**) Roc curve showing the AUC of the Prognostic Index showing the need for biological therapy.

**Table 1 microorganisms-10-01273-t001:** Demographics showing diagnosis, age, sex and fecal calprotectin values.

Demographics	CD	UC	Non-IBD	Healthy
Patients, n (%)	80 (100)	27 (100)	50 (100)	75 (100)
Age, median (range)	13 (0.74–17.9)	11.5 (4–17)	12 (3.7–18)	10 (2–17.9)
Sex (male), n (%)	43 (54)	11 (41)	18 (36)	34 (45)
Fecal calprotectin > 1000 mg/kg, n (%)	31 (39)	12 (48)	2 (4)	0

**Table 2 microorganisms-10-01273-t002:** Bacterial species exhibiting deviating abundances between patient and control groups.

Bacteria	IBD vs. Healthy, (*p*-Value) ^a^	Non-IBD vs. Healthy, (*p*-Value) ^a^	IBD vs. Non-IBD, (*p*-Value) ^a^
**Actinomycetota**			
*Paraprevotella clara*			↓ 0.025
*Bifidobacterium adolescentis*			↓ 0.003
*Bifidobacterium angulatum*	↓ 0.003	↓ 0.032	↓ 0.004
*Bifidobacterium bifidum*	↓ 0.009		↓ 0.004
*Bifidobacterium catenulatum*	↓ 0.000		
*Bifidobacterium longum*	↓ 0.022		
*Bifidobacterium pseudocatenulatum*	↓ 0.004		
**Bacillota**			
*Christensenella minuta*	↓ 0.000		↓ 0.018
*Clostridium leptum*	↓ 0.000		↓ 0.025
*Clostridium scindens*	↓ 0.042		
*Coprococcus comes*		↓ 0.042	
*Desulfovibrio piger*			↓ 0.027
*Eubacterium eligens*	↓ 0.000	↓ 0.000	
*Eubacterium rectale*	↓ 0.001		↓ 0.016
*Eubacterium siraeum*		↓ 0.009	↓ 0.045
*Eubacterium ventriosum*	↓ 0.000		↓ 0.001
*Lactobacillus acidophilus*	↓ 0.038		↓ 0.009
*Lacticaseibacillus paracasei*	↓ 0.001		↓ 0.031
*Methano smithii*	↓ 0.007		
*Parapravotella clara*	↓ 0.003		↓ 0.025
*Roseburia intestinalis*	↓ 0.009		↓ 0.002
*Roseburia inulinivorans*	↓ 0.000		↓ 0.005
*Roseburia hominis*	↓ 0.001		↓ 0.034
*Ruminococcus bromii*	↓ 0.000	↓ 0.014	↓ 0.005
*Streptococcus sanguinis*	↓ 0.031		
*Streptococcus thermophilus*	↓ 0.000	↓ 0.013	
**Bacteroidota**			
*Alistipes finegoldii*	↓ 0.002		↓ 0.002
*Alistipes shahii*	↓ 0.006	↓ 0.006	
*Alistipes onkerdonkii*	↓ 0.001		
*Anaerostipes hadrus*	↓ 0.001		
*Bacteroides caccae*	↓ 0.032	↓ 0.032	
*Bacteroides dorei*	↓ 0.038	↓ 0.043	
*Bacteroides plebeius*			↓ 0.028
*Bacteroides stercoris*			↓ 0.038
*Barnesiella intestinihominis*	↓ 0.004		
**Firmicutes**			
*Intestinibacter bartlettii*	↓ 0.017		
*Enterococcus faecium*		↑ 0.05	
*Citrobacter koseri*		↑ 0.05	
**Pseudomonadota**			
*Haemophilus parainfluenzae*			↑ 0.009
**Verrucomicrobiota**			
*Akkermansia muciniphila*	↓ 0.000		↓ 0.032

^a^ ↑, increase; and ↓, decrease.

**Table 3 microorganisms-10-01273-t003:** Bacterial species exhibiting deviating abundances between medical therapy groups and between groups with high and low calprotectin levels.

Bacteria	Biological vs. Conventional Therapy, (*p*-Value) ^a^	Calprotectin Levels > 1000 mg/kg vs. < 1000 mg/kg, (*p*-Value) ^a^
**Actinomycetota**		
*Bifidobacterium adolescentis*	↓ (0.032)	
*Bifidobacterium bifidum*	↓ (0.044)	
**Bacillota**		
*Eubacterium rectale*	↓ (0.018)	
*Roseburia hominis*	↓ (0.012)	↓ (0.000)
*Roseburia inulinivorans*	↓ (0.011)	↓ (0.028)
*Roseburia intestinales*		↓ (0.001)
*Ruminococcus gnavus*	↑ (0.029)	
*Ruminococcus bromii*	↓ (0.040)	
**Bacteroidota**		
*Alistipes finegoldii*		↓ (0.006)
*Bacteroides finegoldii*	↓ (0.016)	
*Bacteroides intestinalis*	↓ (0.039)	
*Bacteroides vulgatus*	↑ (0.017)	
*Bacteroides cellulosilytius*	↓ (0.035)	
*Barnesiella intestinihominis*		↓ (0.028)
*Paraprevotella clara*	↓ (0.017)	

^a^ ↑, increase; and ↓, decrease.

**Table 4 microorganisms-10-01273-t004:** Bacterial species included in the Diagnostic Indexes 1,2,3, the Phenotype Index and the Prognostic Index.

Bacteria	Diagnostic Index 1 ^a,b^	Diagnostic Index 2 ^a,b^	Diagnostic Index 3 ^a,b^	Phenotype Index ^a,b^	Prognostic Index ^a,b^
**Actinomycetota**					
*Bifidobacterium adolescentis*	1*x		2*x	3*x	
*Bifidobacterium angulatum*	1*x	1*x	1*x		
*Bifidobacterium bifidum*	1*x		3*x	1*x	1*x
**Bacillota**					
*Clostridium leptum*	1*x		1*x		
*Coprococcus comes*		1*x		1*x	
*Eubacterium eligens*	1*x	1*x		1*x	
*Eubacterium rectale*	2*x		1*x		
*Eubacterium ventriosum*	1*x		1*x	1*x	
*Parapravotella clara*	1*x				1*x
*Roseburia intestinalis*	1*x		1*x	1*x	
*Roseburia inulinivorans*	1*x				1*x
*Roseburia hominis*	2*x		1*x		1*x
*Ruminococcus bromii*	1*x				1*x
**Bacteroidota**					
*Alistipes onkerdonkii*	1*x		3*x		
*Alistipes shahii*	1*x	1*x	1*x		
*Alistipes finegoldii*	1*x		1*x	1*x	
*Anaerostipes hadrus*	1*x			1*x	
**Verrucomicrobiota**					
*Akkermansia muciniphila*	2*x		1*x		

^a^ Diagnostic Index 1 = IBD vs. healthy; Diagnostic Index 2 = non-IBD vs. healthy; Diagnostic Index 3 = IBD vs. non-IBD; Phenotype Index = UC vs. CD; Prognostic Index = biologic therapy vs. conventional therapy. ^b^ 1*x, 2*x and 3*x provide the times the bacterial species are weighted in the five indexes.

## Data Availability

The raw datasets generated and analyzed during the current study are not publicly available in order to protect participant confidentiality. The case report forms (CRFs) on paper are safely stored. The data were transferred to SPSS for statistical analyses, and the data files are stored by Akershus University Hospital, Lørenskog, Norway, on a server dedicated to research. The security follows the rules given by The Norwegian Data Protection Authority, P.O. Box 8177 Dep. NO-0034 Oslo, Norway. The data are available on request to the authors.

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
