# Peer review of "Targeted Analysis of the Gut Microbiome for Diagnosis, Prognosis and Treatment Individualization in Pediatric Inflammatory Bowel Disease"

_microorganisms, 2022, doi:10.3390/microorganisms10071273_

Round 1

Reviewer 1 Report

In this study Olbjørn and colleagues present and extensity study of the microbiota profiling for diagnosis, prognosis and treatment individualization in pediatric inflammatory bowel disease.

I found this topic and biomarker research for diagnosis and prognosis very important and of great value for clinicians in the monitoring and management of IBD, especially in patients who may benefit from biological therapy. I appreciate the large cohorts and group with symptoms similar to IBD but without inflammation and the possibility to obtain the results in quite short time.

I have several questions and comments to the authors:

1) In methods the authors were using Mann-Whitney Wilcoxon tests (line 125), this should be probably Mann-Whitney test. Authors declare that they analyzed 110 IBD patients, but in specification of this group, there are 80 patients suffering from UC and 27 patients suffering from UC (the sum is 107 not 110).

2) The gut microbiome in children is known to be dynamic and undergoing development and shaping until the 3rd year of age. The lowest age of infants included into this study was bellow three years of age (CD, HC group). Did the microbiota composition differ inside the group across the age? There are studies showing the differences in microbiota of breast feeding children and children which get formula only, did you observed any differences between these subgroups in your study groups to minimize the effect of diet on your results. Were there differences accoriding the gut involvement in IBD group?

3) In the second part of the study, where authors were analyzing the diagnostic performances, I was missing the simplifying of the indexing for the easy-to-read the data included in the used formula. I strongly suggest to accompany these figre of indexes with graphical representation (e.g. heatmaps) showing the profile of the particular compared group (index). Were any differences between indexes statistically significant? Based on authors observations, could they provide guidelines for clinicians with selected bacteria which was important for the calculation of indexes?

4) Please correct/improve description of graph and axes in figure 3a, 4a () and 3b (violet line, its description); unify the color of box plots (green versus violet color of HC group) and reference line in ROC graphs, use all Figures in same quality (e.g. Fig1b versus Fig2b).

Author Response

See attached file.

Thank you very much for your time, effort and expertise in reviewing our manuscript. We are very thankful for the valuable comments and believe it has been of great benefit for the manuscript. We have taken it all into consideration as you can see in our point by point answers. We have specilfically added a new Table , showing the important bacteria included in the constructed indexes.

Reviewer 2 Report

In this study, Olbjørn et al. investigated the associations between the human gut microbiota and IBD (CD, and UC) using the targeted qPCR approach. Based on the abundance of microbial species, they constructed several IBD-related diagnostic and prognostic indexes. 

Major comments:

1. Authors should explain why they did not choose the state of art methods to quantify microbial community, such as 16S rRNA gene sequencing and whole-genome sequencing.

2. Microbiota analysis: how many species were covered by this targeted qPCR method? what's the primer sequence? 

3. Statistical analyses: Did the author perform the feature selection? And why leave-one-out cross-validation? 

4. For the differential abundance analysis, authors should use some methods that can take care of the confounding factors as listed in table 1: age, sex, and IBD-related treatment. 

5. These IBD-related comparisons are unbalanced classification tasks, and AUC is not a good metric for unbalanced classification. Authors should use AUPRC instead.

6. There are branches of classification models (Random Forest, XG-boost, etc), and the author should compare those models with the logistic regression model.

Minor comments:

1. Title: " Precision Microbiota Profiling" should be "Targeted Microbiota Profiling".

2. All figures: add statistical P-value. 

Author Response

Thank you so much for your valuable time and expertise in reviewing our manuscript. We have replied to your suggestions , see attached file, and believe our manuscript has improved thanks to your helpful comments. We have specifically added a new Table, illustrating the bacteria included in the different Indexes.

Round 2

Reviewer 2 Report

The manuscript has significantly improved and the authors have satisfyingly answered all my comments.